# A Novel Deep Learning Radiomics Model to Discriminate AD, MCI and NC: An Exploratory Study Based on Tau PET Scans from ADNI [note 1]

**DOI:** 10.3390/brainsci12081067

**Published:** 2022-08-12

**Authors:** Yan Zhao, Jieming Zhang, Yue Chen, Jiehui Jiang

**Affiliations:** 1Nuclear Medicine and Molecular Imaging Key Laboratory of Sichuan Province, Luzhou 646000, China; 2Department of Nuclear Medicine, Affiliated Hospital of Southwest Medical University, Luzhou 646000, China; 3Institute of Nuclear Medicine, Southwest Medical University, Luzhou 646000, China; 4School of Pharmacy, Southwest Medical University, Luzhou 646000, China; 5School of Communication and Information Engineering, Shanghai University, Shanghai 200444, China; 6Institute of Biomedical Engineering, School of Life Science, Shanghai University, Shanghai 200444, China

**Keywords:** Alzheimer’s disease, mild cognitive impairment, tau positron emission tomography, deep learning radiomics

## Abstract

Objective: We explored a novel model based on deep learning radiomics (DLR) to differentiate Alzheimer’s disease (AD) patients, mild cognitive impairment (MCI) patients and normal control (NC) subjects. This model was validated in an exploratory study using tau positron emission tomography (tau-PET) scans. Methods: In this study, we selected tau-PET scans from the Alzheimer’s Disease Neuroimaging Initiative database (ADNI), which included a total of 211 NC, 197 MCI, and 117 AD subjects. The dataset was divided into one training/validation group and one separate external group for testing. The proposed DLR model contained the following three steps: (1) pre-training of candidate deep learning models; (2) extraction and selection of DLR features; (3) classification based on support vector machine (SVM). In the comparative experiments, we compared the DLR model with three traditional models, including the SUVR model, traditional radiomics model, and a clinical model. Ten-fold cross-validation was carried out 200 times in the experiments. Results: Compared with other models, the DLR model achieved the best classification performance, with an accuracy of 90.76% ± 2.15% in NC vs. MCI, 88.43% ± 2.32% in MCI vs. AD, and 99.92% ± 0.51% in NC vs. AD. Conclusions: Our proposed DLR model had the potential clinical value to discriminate AD, MCI and NC.

## 1. Introduction

Alzheimer’s disease (AD) is the most prevalent cause of dementia and the most significant disease threatening the health of the elderly [1]. In the early stages of AD, patients often exhibit mild cognitive damage, i.e., mild memory loss mild executive function decrements (e.g., amyloid and tau pathological mechanism), and visuospatial impairment [2,3]. Mild cognitive impairment (MCI) is an intermediate step between normal aging and dementia [4], where patients start to appear memory impairment or other cognitive abnormalities, but have not reached the severity of dementia. Mild cognitive impairment subjects are at high-risk step for dementia [5]. Therefore, it is important to discriminate AD, MCI and normal control (NC) individuals [6,7].

tau positron emission tomography (tau-PET) imaging technology has become increasingly popular for the clinical diagnosis of AD and MCI [8,9,10]. The degree of brain tau accumulation, as an objective biomarker, is strongly correlated with the severity of AD. Johnson et al. found that abnormally high cortical tau binding in the inferior temporal gyrus was associated with clinical impairment [11]. Zhao et al. found that typical deposits of tau appeared in the amygdala, entorhinal cortex, fusiform and parahippocampus in AD brains [12]. La Joie et al. included 28 AD patients and 25 patients with a non-AD clinical neurodegenerative diagnosis and found that tau-PET standard uptake value (SUVR) in the whole brain showed excellent discrimination power (area under curve (AUC) = 0.92–0.94) for diagnosing AD and MCI [13]. Sun et al. proposed a random forest diagnostic model for the classification of NC, MCI and AD and achieved an accuracy of 81.6% [14]. However, existing diagnosis models still have shortcomings, such as the need to manually extract features from the region of interest (ROI) and to encode the extracted features, which often requires tedious processes. Thus, an alternative approach is needed.

Deep learning radiomics (DLR) methods may be the alternative approach. DLR techniques are able to learn high-dimensional features from medical images autonomously and overcome shortcomings such as the cumbersome manual coding in traditional methods [15,16]. In recent years, DLR models have been used in AD studies [17,18]. For instance, Basaia et al. used a deep neural network to classify AD and MCI based on cross-sectional structural resonance imaging (MRI) images. The classification accuracy between AD and NC was 98.2%, and the accuracy of progression from MCI to AD was 74.9% [19]. Lee et al. employed a DLR model for AD classification based on MRI images and achieved an accuracy of 95.35% [20]. Pan et al. proposed a novel convolutional neural network (CNN) architecture called a multi-view separable pyramid network (MiSePyNet) and achieved a classification accuracy of 83.05% in predicting the progression from MCI to AD [21]. Lu et al. used multiscale neural networks to identify subjects with pre-symptomatic AD and achieved an accuracy of 82.51% based on 18F-fluorodeoxyglucose positron emission tomography (FDG-PET) images [22]. The above results showed the feasibility of DLR models for diagnosing AD and MCI. However, whether DLR models could be used to analyze tau-PET images is still unknown. Therefore, in this study, we assumed that the DLR technique was also feasible for application to tau-PET images and would be useful for the diagnosis of AD and MCI. To test the above hypothesis, we employed a novel DLR model and validated it in an exploratory study.

## 2. Methods and Materials

Figure 1 shows the whole experimental process of this study, which includes the following six steps: (1) subject enrollment; (2) tau-PET image preprocessing, including registration, smoothing, and numerical normalization; (3) deep learning (DL) model pre-training. During this session, several classical CNN models were selected and compared, and the best one was finally selected for the next step; (4) extraction of DLR features; (5) classification; (6) comparative experiments.

### 2.1. Subjects

The data used in this study were obtained from the ADNI cohort, which was jointly funded by the National Institutes of Health and the National Institute on Aging in 2004. ADNI is currently the definitive data center for AD-related disease research. In order to obtain the pathogenesis of AD and find treatments, ADNI aims to study the pathogenesis of AD and discover clinical, imaging, genetic and biochemical biomarkers that can be used for the early detection of AD by collecting and organizing longitudinal data from AD patients; the database currently has more than 2000 neuroimaging data. Specific information is available on ADNI’s official website: http://adni.loni.usc.edu/about/, accessed on 12 November 2021.

In this study, a total of 211 NC subjects and 197 MCI and 117 AD patients were collected. All acquired subjects had both T1-weighted MRI images and tau-PET images. Of these, 189 NC subjects and 173 MCI and 101 AD patients were used to train and validate the DLR model. A separate 20 NC subjects and 18 MCI and 12 AD patients were used as an independent external test group. The remaining 2 NC subjects and 6 MCI and 4 AD patients were not included in the training or testing groups because images were found to be mutilated during pre-processing inspections. Demographic information (including gender, age and education) and T1-weighted MRI and tau-PET (AV 1451) images were collected for all participants. All subjects were also screened with the following neuropsychological examinations: the Clinical Dementia Rating-Sum of Boxes (CDR-SB), the MMSE, the MoCA-B, the 11-item and 13-item AD assessment cognitive scale (Alzheimer’s disease assessment scale-cognitive, ADAS) and the ADAS delayed word recall (ADASQ4) subscale. Figure 2 shows the flow chart of the data inclusion/exclusion criteria.

The inclusion criteria of MCI were according to the criteria proposed by Jak and Bondi in 2014 [5]: (1) Scores obtained in at least one cognitive domain (memory, language or speed, executive function) were below the standard deviation of the age/education corrected normative mean; (2) scores in each of the three cognitive domains of memory, language and speed/executive function were found to be impaired; (3) Scores on the Functional Activities Questionnaire (FAQ) ≥ 9. The diagnosis of AD was primarily based on guidelines provided by the National Institute on Aging (NIA) and the Alzheimer’s Association (AA) working group. The ADNI institutional review board reviewed and approved the ADNI data collection protocol [7].

### 2.2. Images Acquisition and Preprocessing

The image acquisition process is described on the ADNI website at http://adni.loni.usc.edu/about/, accessed on 1 June 2021. All tau-PET images were preprocessed using SPM12 software (https://www.fil.ion.ucl.auk/spm/software/spm12/, accessed on 20 September 2021.) implemented in MATLAB 2019b. The preprocessing steps were as follows.

First, the DICOM images were uniformly converted to NIFTI format (.nii) using an image conversion tool for subsequent processing. The converted images were 3D image data with spatial structure information of the brain and retained the characteristic information between tissue structures. Second, since subjects might have some head tilt problems during tau-PET image acquisition, the original correction function in SPM12 was used in this experiment reduce external differences. Furthermore, the T1 MRI images were used to align the tau-PET images so that the corresponding points at spatially uniform locations in the two types of images corresponded to each other. Smoothing and numerical normalization were performed in the next step. After completing the above processing, the images were smoothed to suppress the noise, and the numerical normalization could eliminate the differences between different instruments and reduce the number of subsequent calculations. In this experiment, the images were normalized according to the tau-PET precipitated area in the cerebellar cortex region. After the above processing, 3D image data with a size of 91 × 109 × 91 voxels in the standard space were obtained. To speed up the training time of the DLR models, all images were further normalized to −1 to 1 interval. In the unidirectional slicing condition, the 3D images were axially sliced into 91 single-channel images of size of 91 × 109 voxels, and the slices were filled and resampled to 224 × 224 voxels using linear interpolation due to the need to retain as much information as possible and to satisfy the model input conditions.

### 2.3. The Proposed DLR Model

The proposed DLR model is depicted in Figure 3. The model consists of the following steps: (1) DLR model pre-training. Five classical CNN networks were used for model pre-training. After comparison, we aimed to select the model with the best classification performance; (2) DLR feature extraction and fusion. Based on the pre-trained model, DLR features were extracted before the final maximum pooling layer and combined with clinical features; (3) classifiers: based on the features extracted above, support vector machine (SVM) was employed as the final classifier to obtain the classification results. The details of the model will be illustrated in the next sections.

#### 2.3.1. DLR Model Pre-Training

In recent years, CNN models have been increasingly applied to medical imaging data and shown great potential in the classification tasks. In this study, we pre-trained five common CNN models, including AlexNet, ZF-Net, ResNet18, ResNet34, and InceptionV3 models.

(1)AlexNet demonstrates the excellent performance of deep CNN models. ReLU is used as the activation function for its network structure, which employs interleaved pooling in CNN models [23].(2)ZF-Net is an improved CNN model based on AlexNet. Deconvolution is used to analyze feature behavior and then to improve classification performance [24].(3)The Inception models have more complex network structures and unique network characteristics in comparison with AlexNet and ZF-Net. The Inception structure is designed to use multiple convolutional or pooling operations to form a network module. Inception V3, as the classic version of the Inception series, uses convolutional decomposition and regularization to enhance the classification performance [25].(4)The ResNet framework introduces the residual network structure to solve the gradient disappearance or gradient explosion problem [26]. Different ResNet models, such as ResNet18, ResNet34, ResNet50 and ResNet101, are depending on the number of hidden layers. Figure 4 shows the structures of ResNet18 and ResNet34.

The whole model pre-training process was divided into two parts: forward propagation and backward propagation. Before building the model, all tau-PET images were sliced and tiled into two-dimensional images and adjusted to 224 × 224 pixels. Then, all data were labeled using unique thermal coding. In the model pre-training step, all data were passed into the network and then converged using the stochastic gradient descent (SGD) algorithm and back propagated to update the model parameters. The final output of the model pre-training process was used as the classification result.

In the model pre-training step, we set the learning rate to 1 × 10^−2^ and updated the model parameters using an SGD optimizer with a batch size of 8. The number of training iterations was set to 100. In addition, we performed data enhancement in the training/validation group by flipping the images horizontally and adding Gaussian noise to the input images to prevent the overfitting problem. The above experiments were performed on GPU (graphics processing unit, RTX3090 accelerated by PyCharm 3.6 (JetBrains from the Czech Republic, website: https://www.jetbrains.com/pycharm/, accessed on 17 February 2022)).

#### 2.3.2. DLR Feature Extraction and Fusion

In contrast to traditional methods relying on manually ROI segmentation, DLR methods can automatically leverage tau-PET images to obtain high-dimensional DLR features through supervised learning. After obtaining the best DL pre-trained model, we replaced the final maximum pooling layer and fully connected layer with the SVM as the final classifier. We extracted features from the last convolutional layer of each convolutional network. These features were treated as DLR features. Then, the clinical information (gender, age, education, CDR-SB, MoCA-B, MMSE, etc.) and the DLR features were combined as the input for the SVM classifier.

#### 2.3.3. Classifier

SVM was used as the classifier in this study. SVM is essentially a linear classifier that maximizes intervals in feature space and is a binary classification model that has been widely used with statistical and regression analysis species [27]. We used a linear kernel as the kernel function.

### 2.4. Comparative Experiments

To demonstrate the superiority of the proposed DLR model, we compared the DLR model with three existing models, including: (1) The clinical model. This model includes demographic information and neuropsychological cognitive assessment tests as features for classification; (2) the standard uptake value ratio (SUVR) model. We calculated SUVR values of 10 tau-PET Meta ROIs as features for classification. The ten ROIs included inferior temporal lobe, lingual gyrus, middle temporal lobe, occipital lobe, parietal lobe, hippocampus, parahippocampus, posterior cingu late gyrus, precuneus and fusiform [28]; (3) the radiomics model. The radiomics features of the above 10 tau-PET Meta ROIs were extracted as features for classification. In this experiment, we used the Radiomics Toolkit (https://github.com/mvallieres/radiomics, accessed on 17 February 2022) to extract radiomics features. The feature extraction steps included wavelet band-pass filtering, isotropic resampling, Lloyd–Max quantization, feature computation, and so on [29,30]. In the comparison experiments, each model was with 10-fold cross-validation 200 times.

### 2.5. Statistical Analysis

In this study, chi-square tests and nonparametric rank sum tests were introduced to compare differences in demographic characteristics between the training/validation group and the test group. We used SPSS version 25.0 software (SPSS Inc., Chicago, IL, USA) for all statistical analyses. Statistical results with *p* values < 0.05 were considered significantly different.

## 3. Results

### 3.1. Subject Demographics

Table 1 shows the demographic results. There were differences in MoCA-B (*p* = 0.042) and age (*p* = 0.03) in the NC group; ADAS11 (*p* = 0.020), ADAS13 (*p* = 0.034) and ADASQ4 (*p* = 0.044) in the MCI group and no significant differences in the AD group.

All data are expressed as mean ± standard deviation. MMSE, Mini-mental State Examination; MoCA-B, Montreal cognitive assessment-basic; CDR-SB, clinical dementia rating sum of boxes; ADAS11 and ADAS13, the 11-item and 13-item AD assessment scale cognitive; ADASQ4, the ADAS delayed word recall subscale.

For age, education, MMSE, MoCA-B, CDR-SB, ADAS11, ADAS13 and ADASQ4, a nonparametric rank sum test was performed to compare differences in demographic and clinical characteristics between the training/validation and test groups under each label, i.e., NC, MCI and AD; gender was tested by chi-square between the two groups under each label.

### 3.2. Pre-Training Results of Candidate DL Models

Table 2, Table 3 and Table 4 present the classification performance of the five candidate DL models. The performance evaluation metrics include accuracy, sensitivity and specificity. ResNet18 had the highest classification performance in NC vs. MCI, while ResNet34 had the highest classification performance in MCI vs. AD and NC vs. AD. Therefore, ResNet34 was selected to extract the corresponding DLR features.

### 3.3. Comparative Experiments

#### 3.3.1. NC vs. MCI

Table 5 shows the classification results of the four models in NC vs. MCI. The DLR model performed the best classification performance, with an accuracy of 90.76% ± 2.15%, sensitivity of 94.17% ± 1.81% and specificity of 87.74% ± 2.54% in the test group. The remaining three models performed obviously lower than the DLR model with accuracy of 75.68% ± 2.63%, 72.02% ± 4.12% and 81.61% ± 3.23%; sensitivity of 62.32% ± 4.52%, 68.95% ± 9.22% and 83.11% ± 3.14%; and specificity of 86.67% ± 2.92%, 74.76% ± 7.11% and 80.31% ± 6.38%.

Figure 5 provides the ROC curves of the four models. The AUC (mean ± SD) for the DLR model reached 0.922 ± 0.021 and achieved the best performance among these models.

#### 3.3.2. MCI vs. AD

Table 6 shows the classification performance of the four models in MCI vs. AD. The DLR model showed accuracy of 88.43% ± 2.32%, sensitivity of 91.25% ± 2.05% and specificity of 86.56% ± 2.86% in the test group. The remaining three models performed obviously lower than the DLR model with accuracy of 78.33% ± 4.27%, 79.68% ± 5.72% and 77.16% ± 2.95%; sensitivity of 62.67% ± 9.12%, 65.63% ± 10.97% and 88.17% ± 9.25%; and specificity of 86.67% ± 2.92%, 88.95% ± 2.99% and 68.91% ± 7.64%, respectively.

Figure 6 provided the ROC curves of these four models. The AUC (mean ± SD) for the DLR model reached 0.928 ± 0.024 and achieved the best performance among these models.

#### 3.3.3. NC vs. AD

Table 7 shows the classification performance of the four models in MCI vs. AD. The DLR model showed an accuracy of 99.92% ± 0.51%, sensitivity of 99.78% ± 0.13%, and specificity of 99.99% ± 0.14% in the test group. The remaining three models performed obviously lower than the DLR model with accuracy of 90.66% ± 0.85%, 87.58% ± 3.63% and 96.98% ± 0.21%; sensitivity of 74.96% ± 0.59%, 74.17% ± 9.43% and 92.78% ± 3.13%; and specificity of 99.63% ± 1.33%, 95.24% ± 3.17% and 99.56% ± 2.17%.

Figure 7 provided the ROC curves of these four models. The AUC (mean ± SD) for the DLR model reached 0.996 ± 0.002 and achieved the best performance among these models.

## 4. Discussion

DLR has been becoming a hot topic nowadays. Because of its excellent performance in image recognition and processing, DLR models have been commonly used in computer-aided disease diagnostic fields such lesion detection, quantitative lesion diagnosis, treatment decision and prognosis expectation. In this study, we proposed a DLR model based on tau-PET images to distinguish NC, MCI and AD. In contrast with other traditional models, such as the SUVR model, the traditional radiomics model and the clinical model, the DLR model achieved the best classification results.

To date, many studies have focused on the classification among NC, MCI and AD using machine learning or DL models. For instance, Lange et al. performed a voxel-based statistical analysis using FDG-PET images and achieved an AUC of 0.728 in the classification of AD and NC [31]. Zhou et al. fused MRI and FDG-PET images and used radiomics analysis to achieve an accuracy of 0.733 in the classification of MCI and NC [32]. Shu et al. used radiomics features based on MRI images to classify MCI and AD and achieved an accuracy of 0.807 [33].

Compared with previous studies, our proposed DLR model achieved superior classification results (90.76% ± 2.15% in NC vs. MCI, 88.43% ± 2.32% in MCI vs. AD, and 99.92% ± 0.51% in NC vs. AD). The reasons may be as follows: (1) The DLR model is able to extract deeper image feature information from the pre-processed tau-PET images. As it does not require an additional ROI segmentation step, it decreases errors and biases caused by ROI segmentation; (2) the DLR model is subject to unavoidable external influences such as individual differences and different parameters of imaging acquisition. In our experiment, the DLR features and clinical information were combined together, so bias caused by individual heterogeneities may be eliminated.

Although the DLR model achieved good classification results, several limitations still exist. First, more supporting data are needed to verify the stability of our proposed DLR model. In this research, all data were obtained from the ADNI database. It is worth exploring whether our model works well with other databases. We only used the ADNI database, and the robustness of the results needs to be further verified. In the future, we plan to incorporate other ethnic group data to further verify the effectiveness of our model.

Second, we only adopted five classical deep convolutional networks to obtain the final DLR model. Although the ResNet models performed well in this classification experiment, there may be other more suitable models that can be applied. Moreover, we used whole-brain tau-PET images to train the model, and it remains to be explored whether extracting ROIs would yield better results. In addition to this, in this experiment, the 3D tau-PET images were segmented and sliced according to the axial direction. Whether better results can be obtained using 3D images and convolutional networks at the 3D level requires further validation and experiments. Finally, the model is trained on tau-PET images. Combining with other modalities, such as amyloid PET, MRI and FDG-PET images may improve the classification accuracy. In summary, the DLR model we proposed in this study provides a certain help with the clinical diagnosis and differentiation of NC, MCI and AD. Through this tau-Pet image-based DLR-assisted diagnosis of MCI, early intervention can be carried out for the MCI population, which can improve the cognitive function of patients, allow for early treatment and delay the conversion to dementia.

## 5. Conclusions

In this study, we developed a tau-PET-based DLR method for the subgroup diagnosis of NC, MCI and AD. This study shows that the proposed DLR method can improve the diagnostic performance of MCI and AD patients and provide the possibility of MCI-to-AD conversion prediction. In the future, the DLR method will propose practical applications for the computer-aided diagnosis of MCI and AD. We believe that more image modalities based on our proposed DLR method will be applied in the differential diagnosis of NC, MCI and AD.

## Figures and Tables

**Figure 1 brainsci-12-01067-f001:**
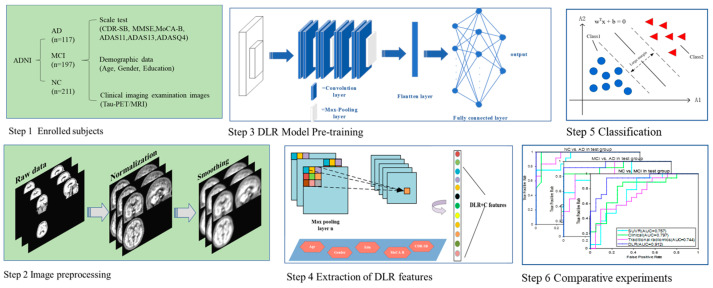
The whole experimental process in this study.

**Figure 2 brainsci-12-01067-f002:**
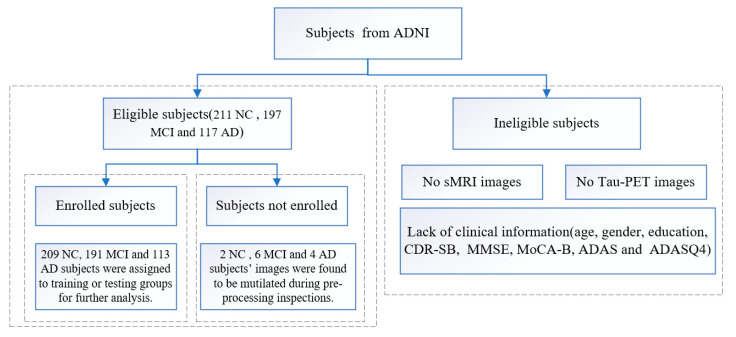
The flow chart of the data inclusion/exclusion criteria.

**Figure 3 brainsci-12-01067-f003:**
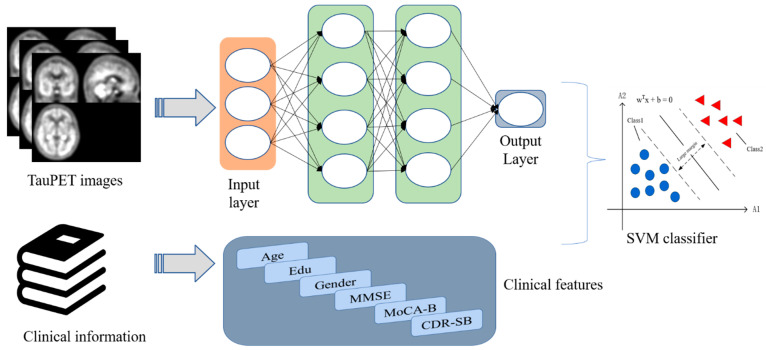
The framework of the proposed DLR model.

**Figure 4 brainsci-12-01067-f004:**
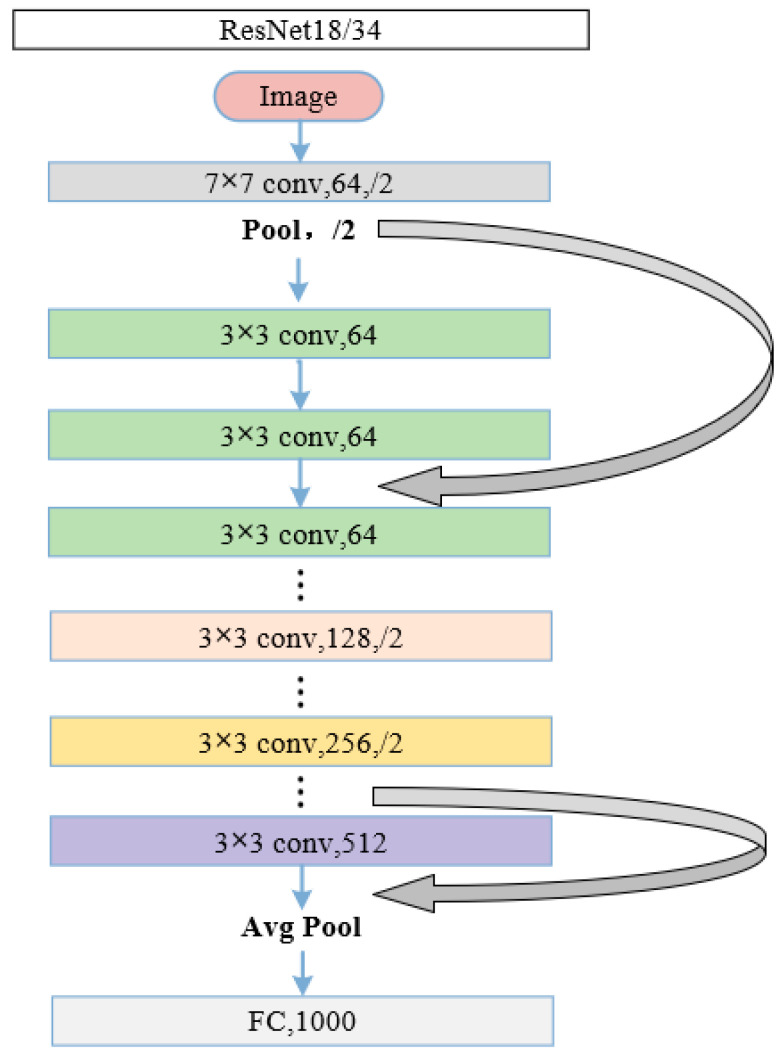
The fundamental structure of ResNet18 and ResNet34. “7 × 7” and “3 × 3” indicate the size of the convolution kernel, “conv” indicates convolution, “Avg Pool” indicates average pooling, and “FC” indicates fully connected layer. “64”, “128”, “128”, “256” and “512” represent the numbers of channels, and “/2” means stride of 2.

**Figure 5 brainsci-12-01067-f005:**
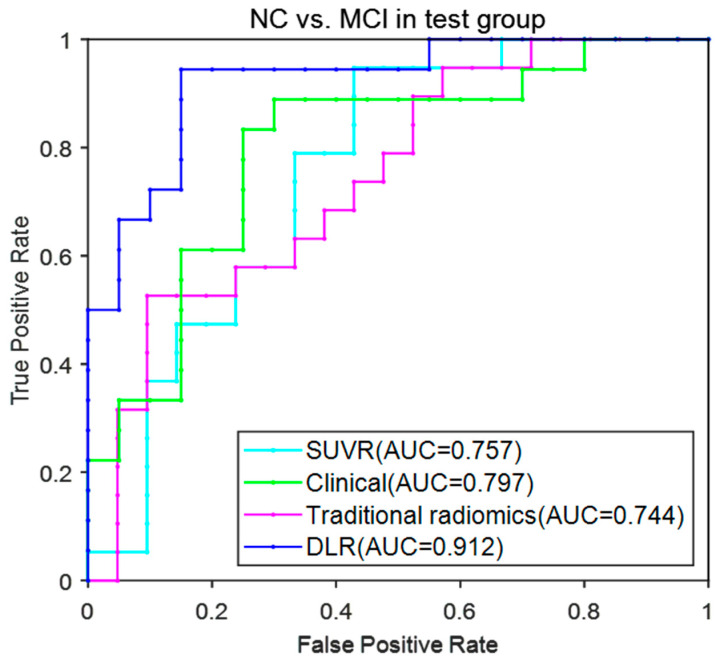
ROC curves for the four models in NC vs. MCI.

**Figure 6 brainsci-12-01067-f006:**
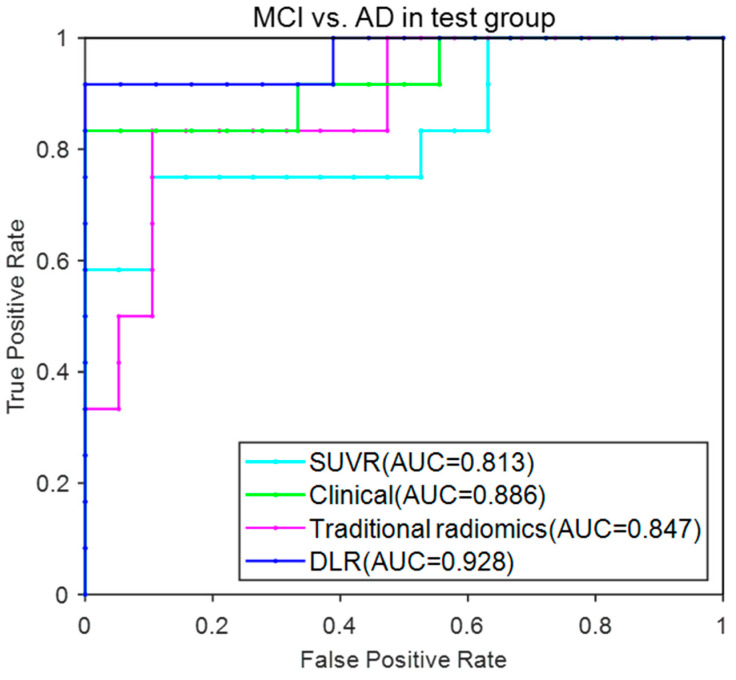
ROC curves for the four models in MCI vs. AD.

**Figure 7 brainsci-12-01067-f007:**
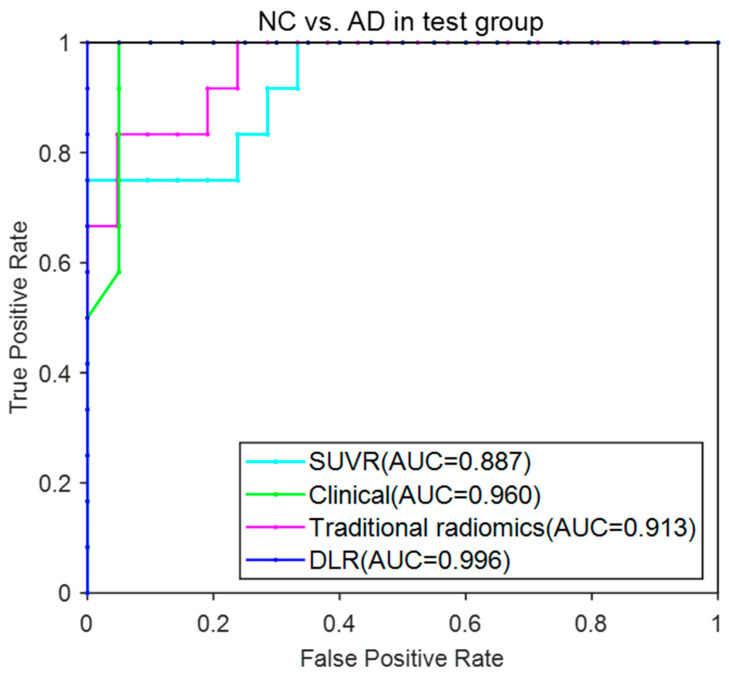
ROC curves for the four models in NC vs. AD.

**Table 1 brainsci-12-01067-t001:** Demographic information in this study.

	NC Groups		MCI Groups		AD Groups	
	NC1 (Train)	NC2 (Test)	MCI1 (Train)	MCI2 (Test)	AD1 (Train)	AD2 (Test)
**Gender (M/F)**	69/121	10/11	95/83	13/6	62/43	5/7
**Age (year)**	73.17 ± 7.64	76.76 ± 6.85 ^a^	73.88 ± 7.46	70.93 ± 8.21	75.39 ± 7.94	76.76 ± 9.38
**Education**	16.66 ± 2.34	17.10 ± 2.04	16.39 ± 2.56	16.05 ± 2.37	15.49 ± 2.59	15.33 ± 2.61
**MMSE**	29.11 ± 1.23	29.14 ± 1.06	27.78 ± 2.21	27.37 ± 2.31	21.36 ± 4.98	21.58 ± 3.55
**MoCA-B**	26.36 ± 2.55	25.00 ± 2.73 ^a^	23.24 ± 3.54	23.68 ± 3.16	16.15 ± 5.03	16.20 ± 4.64
**CDR-SB**	0.06 ± 0.23	0.10 ± 0.20	1.45 ± 1.03	2.37 ± 1.88	5.92 ± 3.32	5.13 ± 2.22
**ADAS11**	8.69 ± 2.57	8.93 ± 1.86	12.59 ± 4.11	14.25 ± 3.08 ^a^	22.10 ± 7.29	25.03 ± 5.98
**ADAS13**	12.46 ± 4.12	13.25 ± 2.99	19.01 ± 6.11	21.51 ± 5.10 ^a^	32.57 ± 8.64	36.11 ± 7.20
**ADASQ4**	2.51 ± 1.72	3.25 ± 2.00	4.77 ± 2.26	5.89 ± 2.35 ^a^	8.06 ± 1.32	8.25 ± 1.76

^a^ indicated that the *p* value was less than 0.05 in comparison results between the training/validation and test groups under the same label.

**Table 2 brainsci-12-01067-t002:** Classification performance in NC vs. MCI.

Model	Accuracy (%)	Sensitivity (%)	Specificity (%)
**Training/Validation Groups**			
AlexNet	94.26 ± 2.60	93.80 ± 2.92	94.70 ± 4.34
ZF-Net	94.28 ± 3.99	94.97 ± 3.64	93.66 ± 7.46
ResNet18	95.78 ± 2.50	94.99 ± 4.70	96.51 ± 2.98
ResNet34	95.32 ± 2.62	94.06 ± 3.74	96.49 ± 4.55
InceptionV3	93.82 ± 3.94	93.02 ± 4.93	94.54 ± 6.39
**Test Group**			
AlexNet	81.25 ± 3.06	7947 ± 2.86	82.86 ± 3.83
ZF-Net	83.14 ± 3.24	78.37 ± 3.89	86.67 ± 5.95
**ResNet18**	**87.25 ± 2.21**	**87.37 ± 2.24**	**87.14 ± 2.32**
ResNet34	87.00 ± 2.14	85.79 ± 2.12	88.10 ± 2.52
InceptionV3	80.50 ± 3.58	77.89 ± 4.24	82.86 ± 5.79

The bold means this model performed best among others.

**Table 3 brainsci-12-01067-t003:** Classification performance in MCI vs. AD.

Model	Accuracy (%)	Sensitivity (%)	Specificity (%)
**Training/Validation Groups**			
AlexNet	93.18 ± 4.36	89.33 ± 10.55	95.41 ± 3.59
ZF-Net	93.55 ± 5.19	91.37 ± 7.20	94.77 ± 5.16
ResNet18	93.72 ± 3.40	90.47 ± 8.16	95.63 ± 4.22
ResNet34	95.28 ± 2.50	94.76 ± 4.96	95.59 ± 3.03
InceptionV3	97.45 ± 2.78	95.26 ± 6.77	98.75 ± 2.64
**Test Group**			
AlexNet	79.68 ± 5.12	64.17 ± 7.32	89.47 ± 4.81
ZF-Net	79.68 ± 2.40	62.50 ± 3.78	90.52 ± 2.34
ResNet18	82.26 ± 1.78	73.33 ± 2.72	87.89 ± 2.14
**ResNet34**	**82.26 ± 1.54**	**77.50 ± 2.48**	**85.26 ± 2.12**
InceptionV3	79.68 ± 2.14	74.17 ± 3.32	83.16 ± 3.48

The bold means this model performed best among others.

**Table 4 brainsci-12-01067-t004:** Classification performance in NC vs. AD.

Model	Accuracy (%)	Sensitivity (%)	Specificity (%)
**Training/Validation Groups**			
AlexNet	97.36 ± 2.98	95.67 ± 7.25	98.27 ± 2.44
ZF-Net	98.30 ± 2.42	97.89 ± 5.09	98.53 ± 2.08
ResNet18	97.17 ± 2.05	96.87 ± 4.42	97.32 ± 2.60
ResNet34	98.10 ± 1.81	96.14 ± 5.87	99.14 ± 1.38
InceptionV3	94.37 ± 3.53	91.29 ± 8.66	96.16 ± 3.13
**Test Group**			
AlexNet	94.24 ± 0.96	84.17 ± 2.63	100.0 ± 0.00
ZF-Net	93.64 ± 2.65	82.57 ± 7.34	100.0 ± 0.00
ResNet18	96.97 ± 2.91	91.70 ± 3.50	100.0 ± 0.00
**ResNet34**	**96.97 ± 2.16**	**91.70 ± 2.83**	**100.0 ± 0.00**
InceptionV3	95.08 ± 3.14	89.58 ± 5.30	98.21 ± 0.96

The bold means this model performed best among others.

**Table 5 brainsci-12-01067-t005:** The classification performance in NC vs. MCI.

Model	Accuracy (%)	Sensitivity (%)	Specificity (%)
**Training/Validation Groups**			
SUVR model	69.36 ± 7.94	63.53 ± 12.03	74.73 ± 10.74
Traditional radiomics model	69.05 ± 7.22	63.34 ± 13.34	74.28 ± 7.07
Clinical model	74.84 ± 8.48	80.13 ± 14.16	71.58 ± 12.05
**DLR model**	**98.46 ± 1.71**	**98.47 ± 1.66**	**98.44 ± 1.76**
**Test Group**			
SUVR model	75.68 ± 2.63	62.32 ± 4.52	86.67 ± 2.92
Traditional radiomics model	72.02 ± 4.12	68.95 ± 9.22	74.76 ± 7.11
Clinical model	81.61 ± 3.23	83.11 ± 3.14	80.31 ± 6.38
**DLR model**	**90.76 ± 2.15**	**94.74 ± 1. 81**	**87.74 ± 2.54**

The bold means this model performed best among others.

**Table 6 brainsci-12-01067-t006:** The classification performance in MCI vs. AD.

Model	Accuracy (%)	Sensitivity (%)	Specificity (%)
**Training/Validation Groups**			
SUVR model	74.41 ± 8.15	55.39 ± 15.64	86.06 ± 8.88
Traditional radiomics model	70.20 ± 7.83	57.79 ± 13.64	81.58 ± 9.83
Clinical model	90.84 ± 4.95	84.59 ± 11.04	94.45 ± 5.00
**DLR model**	**96.27 ± 1.16**	**94.90 ± 4.81**	**97.89 ± 2.11**
**Test Group**			
SUVR model	78.33 ± 4.27	62.67 ± 9.12	86.67 ± 2.92
Traditional radiomics model	79.68 ± 5.72	65.63 ± 10.97	88.95 ± 2.99
Clinical model	77.16 ± 2.95	88.17 ± 9.25	68.91 ± 7.64
**DLR model**	**88.43 ± 2.32**	**91.25 ± 2.05**	**86.56 ± 2.86**

The bold means this model performed best among others.

**Table 7 brainsci-12-01067-t007:** The classification performance in NC vs. AD.

Model	Accuracy (%)	Sensitivity (%)	Specificity (%)
**Training/Validation Groups**			
SUVR model	86.06 ± 6.18	73.43 ± 14.21	93.04 ± 6.06
Traditional radiomics model	78.65 ± 0.08	57.67 ± 16.64	87.06 ± 8.80
Clinical model	91.96 ± 5.44	99.06 ± 2.98	86.65 ± 8.04
**DLR model**	**99.31 ± 1.50**	**98.00 ±4.43**	**100.0 ± 0.00**
**Test Group**			
SUVR model	90.66 ± 0.85	74.96 ± 0.59	99.63 ± 1.33
Traditional radiomics model	85.58 ± 3.63	74.17 ± 9.43	95.24 ± 3.17
Clinical model	96.98 ± 0.21	92.78 ± 3.13	99.56 ± 2.17
**DLR model**	**99.92 ± 0.51**	**99.78 ± 0. 13**	**99.99 ± 0.14**

The bold means this model performed best among others.

## Data Availability

The subject data used in this study were obtained from the Alzheimer’s Disease Neuroimaging Initiative (ADNI) database (adni.loni.usc.edu). Meanwhile, data supporting the findings of this study are available from the corresponding authors upon reasonable request.

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
