# Peer review of "A Novel Deep Learning Radiomics Model to Discriminate AD, MCI and NC: An Exploratory Study Based on Tau PET Scans from ADNI†"

_brainsci, 2022, doi:10.3390/brainsci12081067_

Round 1
Reviewer 1 Report
I reviewed with great interest the article submitted by Jiehui Jiang and co-workers entitled "A novel deep learning radiomics model to discriminate AD, MCI and NC: an exploratory study based on Tau PET scans".
The study is innovative in the field of AI application on brain PET imaging and of potential clinical interest in the differential diagnosis of neurodegenerative disease.
I have some suggestions to improve the article:
- The title should reflect completely the content of the study. I suggest to define the use of ADNI database in the title.
- Despite the authors have considered patients from a standardized and well organised database, the inclusion/exclusion criteria should be defined. A Stard Flow diagram may help the reader;
- The discussion should report other potential limitations and lack of reproducibility in the clinical setting.
- The potential impact on differential diagnosis have to be discussed in a more detailed form;
- The conclusion might be refined. This form is redundant.
Reviewer 2 Report
The authors presented the model for distinguish patients with Alzheimer's disease, mild cognitive impairments as well as normal controls based on Tau-PET scans using deep learning. The model was validated and tested and presented the better results compared to other models. The work is of high importance for diagnostic of Alzheimer's disease and especially for differentiating Alzheimer's disease's and mild cognitive impairments' conditions.
However, there are several comments for authors to improve the manuscript:
1. The authors declare a total of 211 NC subjects, 197 MCI and 117 AD patients; whithin them 189 NC subjects, 173 MCI and 101 AD patients were used to train and validate the model, whereas 20 NC subjects, 18 MCI and 12 AD patients were used to test the model. Summarazing, the authors used 209 NC subjects, 191 MCI and 113 AD patients for training, validation and testing. The authors should add exlusion criteria for 2 NC subjects, 6 MCI and 4 AD patients in the text.
2. Usually titles of tables are placed above tables. Thus, please, put the title of the Table 1 above the table.
3. The authors just copied the text from Instructions for authors instead of the Data Availability Statement. Thus, the authors should add this statement for their data.
